# Phenoxyalkylimidazoles with an oxadiazole moiety are subject to efflux in *Mycobacterium tuberculosis*

Mai B. Thayer[1][¤], Tanya Parish[1,2]*

1 TB Discovery Research, Infectious Disease Research Institute, Seattle, WA, United States of America,
2 Center for Global Infectious Disease Research, Seattle Children's Research Institute, Seattle, WA, United States of America

¤ Current address: Pharmacokinetics and Drug Metabolism Department, Amgen Research, South San Francisco, CA, United States of America
* tanya.parish@seattlechildrens.org

**Data Availability Statement:** All relevant data are within the paper.

**Funding:** TP - Bill & Melinda Gates Foundation under grant OPP1024038 https://www.gatesfoundation.org/ TP - NIAID of the National

## Abstract

The phenoxyalkylimidazoles (PAI) are an attractive chemical series with potent anti-tubercular activity targeting *Mycobacterium tuberculosis* respiration. Our aim was to determine if the PAI compounds are subject to efflux. Two analogs containing an oxadiazole had improved potency in the presence of the efflux inhibitors reserpine and carbonyl cyanide m-chlorophenylhydrazine, whereas the potency of analogs with a diazole was not affected.

## Introduction

Tuberculosis remains a major global health challenge with ~1.5 million deaths and >10 million new cases annually [1]. There is an urgent need for new drugs which can both shorten therapy and treat drug resistant organisms [2, 3]. However, the causative agent *Mycobacterium tuberculosis* is resistant to many antibacterial molecules [4]. This intrinsic resistance has long been thought to be largely as a result of its impenetrable cell wall including a hydrophobic layer of mycolic acids [4]. In addition, efflux and metabolism play a role in reducing molecule efficacy [5].

We investigated the role of efflux in the sensitivity of *M. tuberculosis* to members of the phenoxyalkylimidazole (PAI) series [6, 7] (Fig 1). Our aim was to determine whether any of the compounds were subject to efflux and if this affected their activity. We had previously identified these as being active against the commonly used laboratory strain of H37Rv (ATCC 25618) [6, 7]. The PAI series targets QcrB, the cytochrome b subunit of cytochrome bc1 complex [6, 8]. This complex is part of the electron transport chain which is involved in generating the proton gradient that drives ATP production via ATP synthase [9].

## Methods

### Bacterial culture and strains

*M. tuberculosis* was cultured in Middlebrook 7H9 medium containing 10% v/v oleic acid, albumin, dextrose, catalase (OADC) supplement and 0.05% w/v Tween 80. Strains used were *M.*

Institutes of Health under award number R01AI099188 https://www.niaid.nih.gov/. The funders had no role in study design, data collection and analysis, decision to publish, or preparation of the manuscript.

**Competing interests:** The authors have declared that no competing interests exist.

*tuberculosis* H37Rv (London Pride: ATCC 25618) [10], *M. tuberculosis* Rv0678$_{I67S}$ [11], CDC1551 (BEI Resources) and CDC1551 MmpL5::Tn (BEI Resources).

### Minimum inhibitory concentration

Minimum inhibitory concentrations (MICs) were determined as described [12]. Briefly, bacterial growth was measured after 5 days in medium the presence of a ten point, two-fold dilution series of each compound. Growth was normalized to controls and the MIC$_{90}$ determined as the concentration required to inhibit growth by 90%. MIC$_{90}$s were determined in three independent experiments and reported as the mean ± SD.

### Results and discussion

We wanted to determine if efflux played a role in compound activity using efflux inhibitors. We first determined the concentration of efflux inhibitors to use by running a dose response and selecting the highest concentration at which *M. tuberculosis* growth was not affected: 12.5 μM for reserpine, 20 μM for verapamil and 2.5 μM for carbonyl cyanide m-chlorophenylhydrazine (CCCP). We tested the activity of two members of the PAI series in the presence of these efflux inhibitors (Fig 2). MIC$_{90}$s were determined in three independent experiments and reported as the mean ± SD. Differences of >2-fold were considered as shifts if p<0.05 using Student's unpaired t-test (Fig 2).

Reserpine increased the activity of the two compounds against the wild-type strain (Fig 2A and 2B). For PAI compound 351551, there was a 5-fold increase in potency in the presence of reserpine (p = 0.0004). CCCP also boosted activity 3.4-fold. For PAI compound 390229, there were smaller changes, with a 2.5-fold change with reserpine (p = 0.003) and a 2-fold change with CCCP (p = 0.003). There was no significant difference (p>0.05) in the presence of verapamil for either compound, although the concentration we used was relatively low. We selected a low concentration of verapamil to avoid any effect on growth, as we had previously seen the

**Fig 1. Structure of molecules used in this study.**

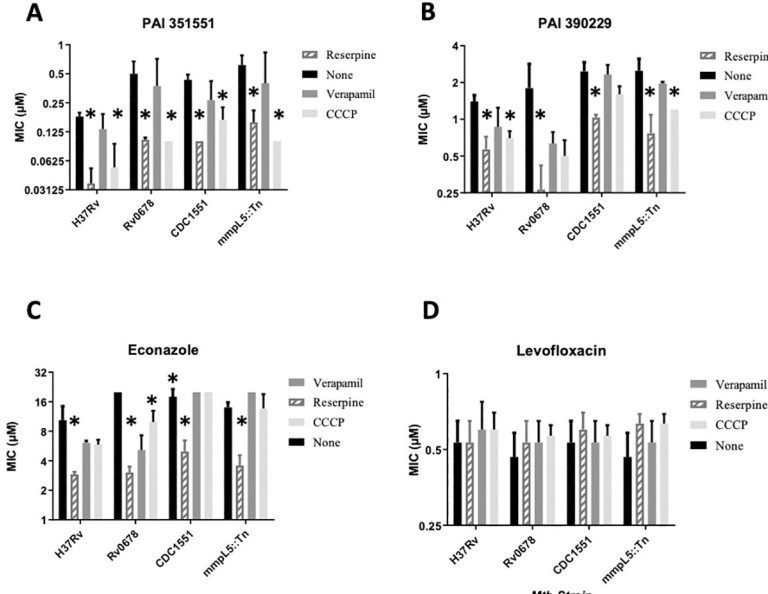

**Fig 2. Activity of PAI compounds in the presence of efflux inhibitors.** $MIC_{90}$s were determined from three independent experiments. Strains tested were H37Rv (ATCC25618), Rv0678 (carrying allele with I67S), CDC1551 wild-type, and MmpL5::Tn (CDC1551 background). Panels: A–PAI 351551. B–PAI 390299. C- Econazole. D– Levofloxacin. Data are mean ± SD. * p<0.05 using Student's unpaired t-test.

$IC_{50}$ as low as 100 μM depending on the batch used. However, it is possible that this concentration was too low to affect efflux and that higher concentrations of verapamil might be required to see any effect. We included econazole and levofloxacin as positive and negative controls respectively (Fig 2C and 2D). Econazole is known to be subject to efflux in *M. tuberculosis* [13]. We confirmed that econazole activity was also increased in the presence of reserpine 3.5-fold (p = 0.03), but not significantly increased by CCCP. Levofloxacin activity was not affected by any efflux inhibitor.

Previous work had demonstrated that econazole efflux was mediated by the MmpL5-MmpS5 system under the negative control of the Rv0678 transcriptional regulator [13]. Inactivation of this regulator leads to upregulation of the efflux system and mediates low level resistance to econazole [13]. We had shown a similar phenotype using compound 105909 (Fig 1) and isolated a strain of *M. tuberculosis* with a SNP in Rv0678 [11]. We tested molecule activity against the strain carrying $Rv0678_{I67S}$ (Fig 2). The strain showed low level resistance to compound 351551, but not to compound 390229 (Fig 2A and 2B). For compound PAI compound 351551, there was a 2.8-fold decrease in potency against the Rv0678 mutant strain (p = 0.03). The addition of reserpine boosted compound activity 5-fold in this background, although it did not reduce fully to wild-type levels (p = 0.02). In this background CCCP also boosted activity to a similar level as reserpine (p = 0.02). For compound PAI compound 390229, although there was no resistance, reserpine was still active; verapamil and CCCP were effective at increasing potency. The Rv0678 mutant was resistant to econazole as expected ($MIC_{90}$ >20 μM) and its activity was boosted by all three efflux inhibitors to varying degrees, with reserpine being the most effective (Fig 2C). There was no difference in activity with the Rv0678 for levofloxacin under any condition (Fig 2D).

In order to investigate the role of the MmpL5-MmpS5 system further, we used a strain in which the system was interrupted by a transposon (MmpL5::Tn). The latter strain was compared to its parental strain CDC1551. There was no difference in $MIC_{90}$ between the wild-type

and Tn mutant strain for PAI compounds, econazole or levofloxacin. Reserpine boosted the potency of both PAI compounds and econazole in both strains, suggesting that this system is not the only mechanism of efflux for econazole. CCCP boosted the activity of the PAI compounds in the MmpL5::Tn mutant strain, but had little to no effect in the CDC1551 wild-type strains. Taken together these data suggest that the PAI compounds are exported by more than one system, and that inhibition of efflux by reserpine is most effective in boosting their potency. Verapamil had no effect on compound activity.

The two compounds we selected for testing from the PAI series both had oxadiazole moieties (Fig 1). Since 105909 also has an oxazole, we wondered if this might be an indicator of efflux. We tested two additional compounds against these strains and under the same conditions (Fig 3). PAI compound 333819 has a diazole moiety instead of an oxadiazole function. The imidazo-pyrimidine compound 106909 is from a different chemical scaffold, but with the same target (QcrB) [12, 14, 15]. We also included the 105909 compound containing an oxazole.

The potency of the IMP or PAI compounds was not improved by the addition of efflux inhibitors in any of the strains (less than 2-fold change in all conditions). For the oxazole compound, there was little change in the activity in the wild-type strain, but in the context of the Rv0678 mutant which had a slightly higher baseline $MIC_{90}$, reserpine was able to boost activity back to wild-type levels (6-fold reduction, p = 0.03). We conclude that the oxazole is only subject to minimal efflux.

We tested an additional four compounds from the PAI series with the diazole moiety (Fig 4). There were no significant differences in $MIC_{90}$ in the presence of any of the efflux inhibitors (<2-fold change). These data suggest that efflux does not occur with all molecules in the PAI series, but only with a subset.

Our data show that two molecules from the PAI series are susceptible to efflux. These data are important from a drug discovery perspective when developing agents based on whole cell activity. The activity of compounds in whole cell assays is dependent on a number of factors including transport uptake efficiency, efflux processes, compound metabolism (activation/inactivation) and ligand-binding efficiency. Our data demonstrate that efflux can affect the activity of some compounds in the PAI series, and therefore that any investigation of structure-activity relationships based on MICs needs to take this into account. Although we cannot make any conclusions about the specificity of the efflux systems for this series, further studies to look at the kinetics of compound uptake and a wider variety of compounds might shed light on whether there are specific chemical moieties which make compounds subject to efflux. Although the differences are small, they are significant and are large enough to complicate analysis of SAR and molecule design for drug discovery. Therefore, efflux should be considered in any further development of the PAI series as anti-tubercular agents.

Reserpine and CCCP both affect efflux by targeting ATP generation, suggesting that efflux of these molecules is an active process. The fact that structurally similar compounds were not affected equally by efflux points to the difficulty of developing chemical series in a rational fashion, since activity is always sum of uptake, efflux, metabolism and target binding. Therefore, changes in activity can result from any of those parameters. In addition, our data show that concerns over efflux are also relevant to compounds with membrane-located targets such as QcrB [9].

## Acknowledgments

We thank Juliane Ollinger, Susantha Chandrasekara, Aaron Korkegian and Theresa O'Malley for discussion and technical assistance.

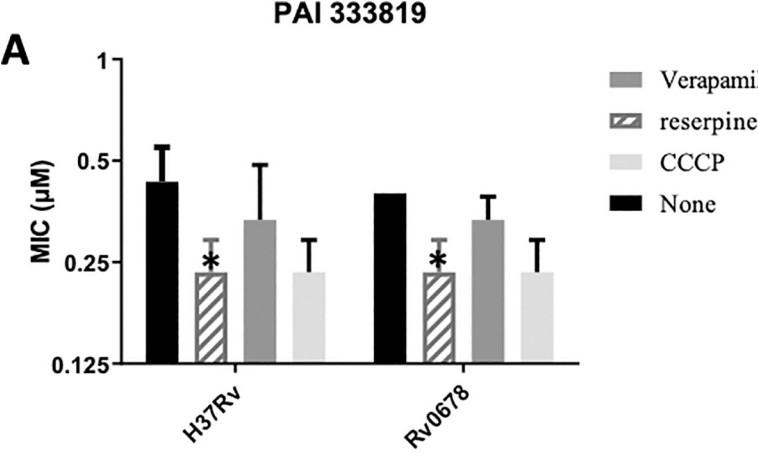

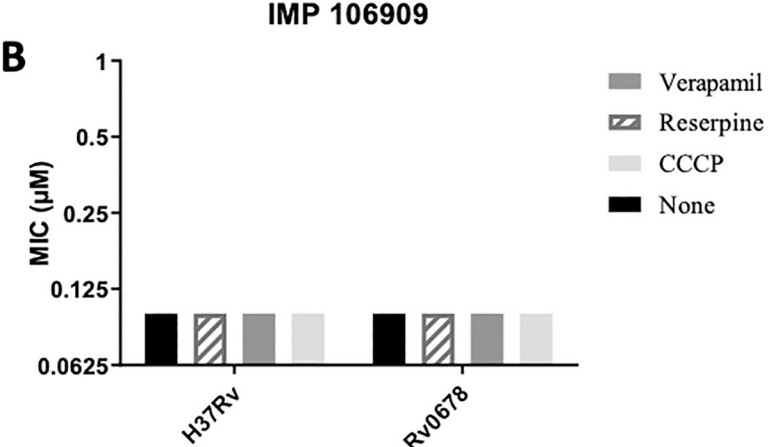

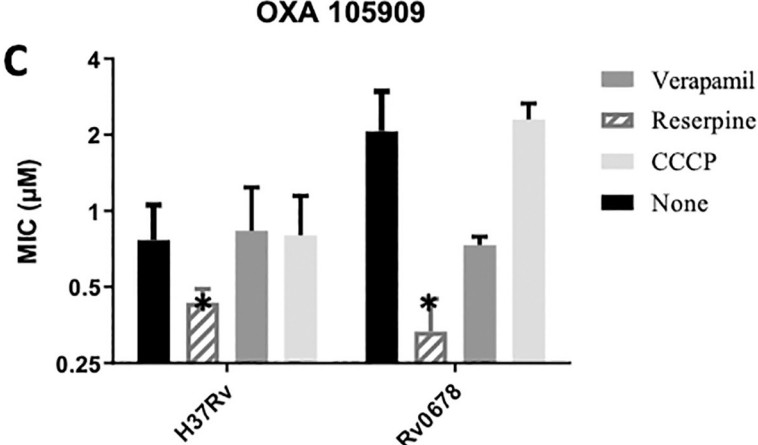

**Fig 3. Activity of anti-tubercular agents in the presence of efflux inhibitors.** MIC$_{90}$s were determined from three independent experiments. Strains tested were H37Rv (ATCC25618) and Rv0678 (carrying allele with I67S). Panels: A–PAI 333819. B–IMP 106909. C—OXA 105909. Data are mean ± SD. * p<0.05 using Student's unpaired t-test.

MIC$_{90}$ (µM)

| Molecule | Structure | None | Verapamil | CCCP | Reserpine |
|---|---|---|---|---|---|
| 458745 | | 2.0 | 1.7 | 1.9 | 3.2 |
| 357430 | | 0.14 | 0.16 | 0.14 | 0.11 |
| 351605 | | 4.1 | 3.3 | 4.4 | 2.8 |
| 351604 | | 9.0 | 8.9 | 7.4 | 5.0 |

**Fig 4. Activity of PAI series in the presence of efflux inhibitors.** The MIC$_{90}$ was determined against *M. tuberculosis* H37Rv [12].

## Author Contributions

**Conceptualization:** Mai B. Thayer, Tanya Parish.

**Data curation:** Mai B. Thayer, Tanya Parish.

**Formal analysis:** Mai B. Thayer, Tanya Parish.

**Funding acquisition:** Tanya Parish.

**Investigation:** Mai B. Thayer, Tanya Parish.

**Methodology:** Mai B. Thayer, Tanya Parish.

**Project administration:** Tanya Parish.

**Resources:** Tanya Parish.

**Supervision:** Tanya Parish.

**Validation:** Tanya Parish.

**Writing – original draft:** Tanya Parish.

**Writing – review & editing:** Mai B. Thayer, Tanya Parish.

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
