## [Decision Letter · Decision Letter 0]

7 Oct 2020

PONE-D-20-28352

Phenoxyalkylimidazoles with an oxadiazole moiety are subject to efflux in Mycobacterium tuberculosis

PLOS ONE

Dear Dr. Parish,

Thank you for submitting your manuscript to PLOS ONE. After careful consideration, we feel that it has merit but does not meet PLOS ONE’s publication criteria as it currently stands.

As noted below, both referees expressed significant concerns about the study, and both considered it to be rather preliminary in its current state. If you feel that you can convincingly address the reviewers' concerns, we invite you to submit a thoroughly revised version of the manuscript.

We look forward to receiving your revised manuscript.

Kind regards,

Olivier Neyrolles, PhD

Section Editor

PLOS ONE

We note that one or more of the authors are employed by a commercial company: Amgen.

2.1. Please provide an amended Funding Statement declaring this commercial affiliation, as well as a statement regarding the Role of Funders in your study. If the funding organization did not play a role in the study design, data collection and analysis, decision to publish, or preparation of the manuscript and only provided financial support in the form of authors' salaries and/or research materials, please review your statements relating to the author contributions, and ensure you have specifically and accurately indicated the role(s) that these authors had in your study. You can update author roles in the Author Contributions section of the online submission form.

2.2. Please also provide an updated Competing Interests Statement declaring this commercial affiliation along with any other relevant declarations relating to employment, consultancy, patents, products in development, or marketed products, etc.  

Reviewers' comments:

Reviewer's Responses to Questions

**Comments to the Author**

1. Is the manuscript technically sound, and do the data support the conclusions?

Reviewer #1: No

Reviewer #2: No

2. Has the statistical analysis been performed appropriately and rigorously? 

Reviewer #1: Yes

Reviewer #2: Yes

3. Have the authors made all data underlying the findings in their manuscript fully available?

Reviewer #1: Yes

Reviewer #2: Yes

4. Is the manuscript presented in an intelligible fashion and written in standard English?

Reviewer #1: Yes

Reviewer #2: Yes

5. Review Comments to the Author

Reviewer #1: In previous publications, the antituberculosis activity of PAI compounds has been fully described, and now, in the present manuscript, the impact of efflux on their activity has been investigated.

In this work, it is reported that verapamil, a widely used and very potent efflux inhibitor in mycobacteria, has failed to decrease MICs of the PAI compounds. A potential explanation could be that in this work, verapamil is used at 20 µM whereas in most publications on the field verapamil is used at 180 µM (or even higher concentrations!), without compromising cell viability. Maybe, such a difference in concentration could explain this discrepancy. It is strongly suggested to further test this concentration experimentally.

The relationship between presence of oxadiazole moieties and efflux susceptibility has been explored further by analysing a few other compounds. First, structures shown in Figure 1 could maybe be arranged in a different way in order to make easier the identification of structural similarities and differences between them. Second, I agree with the authors’ perception (shown in the last paragraph of the manuscript) that a much wider number of compounds should be assayed in order to fully support (or not) the influence of this chemical part in efflux susceptibility.

Minor points:

Typos: In methods section, albumen -> albumin; second line in MIC paragraph;

First four lines in Results and Discussion could be moved to introduction

Description of MIC protocol is duplicated in Methods and in Results and Discussion.

It would be good to read a brief description of the activity of QcrB protein.

Reviewer #2: This work examines the effect of efflux inhibitors on a couple of compounds. Two of the compounds have an oxadiazole belonging to the PAI series, one an oxazole and one is an indole-containing PAI compound. Some control compounds are included. The efflux inhibitors have different effects on the MIC of the compounds. The MIC of two compounds (351551 and 105909) are affected by a Rv0678 mutation although a transposon mutant in mmpL5 shows no difference in MIC. From the limited series of compounds explored it seems hard to make structure-associated conclusions about similarities in efflux although reserpine affects the MIC of several compounds.

Major comments

A much broader series of compounds need to be explored if any conclusions are to be made regarding the effect of a chemical moiety on efflux.

Minor comments

The inclusion of MIC testing against the Rv0678 mutant strains and mmpL5 inactivation strain adds data but little clarity. The mutations in Rv0678 confer a small increase in resistance whereas inactivation of mmpL5 gives very little difference overall for most compounds to the MIC compared to the parental strain. For example, for 351551 the Rv0678 mutant has a 2-fold higher MIC but the mmpL5 Tn mutant is not more sensitive than its parental strain. The other compounds are not affected by the Rv0678 mutation.

From just two compounds containing an oxadiazole, it seems hard to make any substantial conclusion about similarities in efflux. The MIC of one is affected by the Rv0678 mutation and the other one hardly at all. Reserpine affects the MIC of both but it also affects the MIC of the oxazole and the non-oxadiazole-containing PAI compound - it may be less than 2-fold, but all the differences are modest yet significant.

6. PLOS authors have the option to publish the peer review history of their article (what does this mean?). If published, this will include your full peer review and any attached files.

Reviewer #1: No

Reviewer #2: No

---

## [Author Response · Author response to Decision Letter 0]

16 Oct 2020

Response as noted in the rebuttal file.

Reviewer #1: 

In previous publications, the antituberculosis activity of PAI compounds has been fully described, and now, in the present manuscript, the impact of efflux on their activity has been investigated.

In this work, it is reported that verapamil, a widely used and very potent efflux inhibitor in mycobacteria, has failed to decrease MICs of the PAI compounds. A potential explanation could be that in this work, verapamil is used at 20 µM whereas in most publications on the field verapamil is used at 180 µM (or even higher concentrations!), without compromising cell viability. Maybe, such a difference in concentration could explain this discrepancy. It is strongly suggested to further test this concentration experimentally.

>As noted in the manuscript, we used the highest concentration of verapamil at which growth was unaffected in our assays. Concentrations higher than 20 µM resulted in growth inhibition with our strain and in our assay. Since higher concentrations affected growth, we conclude that 20 µM should have an effect on efflux. We cannot comment on how other groups were able to use verapamil at higher concentrations without compromising growth. From the manuscript:

“We first determined the concentration of efflux inhibitors to use by running a dose response and selecting the highest concentration at which M. tuberculosis growth was not affected: 12.5 �M for reserpine, 20 �M for verapamil and 2.5 �M for carbonyl cyanide m-chlorophenylhydrazine (CCCP).”

The relationship between presence of oxadiazole moieties and efflux susceptibility has been explored further by analysing a few other compounds. First, structures shown in Figure 1 could maybe be arranged in a different way in order to make easier the identification of structural similarities and differences between them. Second, I agree with the authors’ perception (shown in the last paragraph of the manuscript) that a much wider number of compounds should be assayed in order to fully support (or not) the influence of this chemical part in efflux susceptibility.

>We have revised Figure 1.

>We agree that a thorough study of a wide range of chemical moieties might be able to determine the determinants of efflux, but we feel this is beyond the scope of this manuscript. We have clarified and expanded our rationale for conducting this study. The data generated are important, not for understanding efflux per se, but for guidance in a drug discovery program looking at structure-activity relationships using whole cell activity as the only driver of medicinal chemistry. We have demonstrated that efflux plays a small, but significant role in the activity of two compounds from the PAI series. We have included data from four additional compounds with the diazole moiety which were not subject to efflux, which strengthens our conclusion (Table 1). These data suggest that future development of this series should also consider whether compounds are subject to efflux, especially when looking at structure-activity-relationships. The conclusions are relevant for drug discovery and compound progression. We have tempered and clarified our conclusions to reflect this. 

Minor points:

Typos: In methods section, albumen -> albumin; second line in MIC paragraph;

>Corrected

First four lines in Results and Discussion could be moved to introduction

> We have moved as suggested.

Description of MIC protocol is duplicated in Methods and in Results and Discussion.

>We have deleted from the results section.

It would be good to read a brief description of the activity of QcrB protein.

>We have added this.

Reviewer #2: 

This work examines the effect of efflux inhibitors on a couple of compounds. Two of the compounds have an oxadiazole belonging to the PAI series, one an oxazole and one is an indole-containing PAI compound. Some control compounds are included. The efflux inhibitors have different effects on the MIC of the compounds. The MIC of two compounds (351551 and 105909) are affected by a Rv0678 mutation although a transposon mutant in mmpL5 shows no difference in MIC. From the limited series of compounds explored it seems hard to make structure-associated conclusions about similarities in efflux although reserpine affects the MIC of several compounds.

Major comments

A much broader series of compounds need to be explored if any conclusions are to be made regarding the effect of a chemical moiety on efflux.

>See point for reviewer 1. We have clarified the rationale for this study as supporting a drug discovery effort and included additional data.

Minor comments

The inclusion of MIC testing against the Rv0678 mutant strains and mmpL5 inactivation strain adds data but little clarity. The mutations in Rv0678 confer a small increase in resistance whereas inactivation of mmpL5 gives very little difference overall for most compounds to the MIC compared to the parental strain. For example, for 351551 the Rv0678 mutant has a 2-fold higher MIC but the mmpL5 Tn mutant is not more sensitive than its parental strain. The other compounds are not affected by the Rv0678 mutation.

From just two compounds containing an oxadiazole, it seems hard to make any substantial conclusion about similarities in efflux. The MIC of one is affected by the Rv0678 mutation and the other one hardly at all. Reserpine affects the MIC of both but it also affects the MIC of the oxazole and the non-oxadiazole-containing PAI compound - it may be less than 2-fold, but all the differences are modest yet significant.

>We agree that the differences are small, but the data are robust. We think the information is important to present in the publication but can move to supplementary information if required.

---

## [Decision Letter · Decision Letter 1]

11 Nov 2020

PONE-D-20-28352R1

Phenoxyalkylimidazoles with an oxadiazole moiety are subject to efflux in Mycobacterium tuberculosis

PLOS ONE

Dear Dr. Parish,

Thank you for submitting your revised manuscript to PLOS ONE. After careful consideration, we feel that it has merit but does not fully meet PLOS ONE’s publication criteria as it currently stands. Therefore, we invite you to submit a further revised version of the manuscript that addresses the point raised by Reviewer #1 during the second review process, i.e. **the concentration of verapamil used in this study, compared to in the other studies cited by the Referee, must be properly discussed**.

We look forward to receiving your revised manuscript.

Kind regards,

Olivier Neyrolles

Academic Editor

PLOS ONE

Reviewers' comments:

Reviewer's Responses to Questions

**Comments to the Author**

1. If the authors have adequately addressed your comments raised in a previous round of review and you feel that this manuscript is now acceptable for publication, you may indicate that here to bypass the “Comments to the Author” section, enter your conflict of interest statement in the “Confidential to Editor” section, and submit your "Accept" recommendation.

Reviewer #1: (No Response)

2. Is the manuscript technically sound, and do the data support the conclusions?

Reviewer #1: Partly

3. Has the statistical analysis been performed appropriately and rigorously? 

Reviewer #1: N/A

4. Have the authors made all data underlying the findings in their manuscript fully available?

Reviewer #1: Yes

5. Is the manuscript presented in an intelligible fashion and written in standard English?

Reviewer #1: Yes

6. Review Comments to the Author

Reviewer #1: On the original manuscript, it was raised the question on the concentration of verapamil used for the MIC assays (20µM), which was regarded as too low. Publications from Viveiros´s lab (for example, PMID: 28496433) have reported that MIC of verapamil is 560 µM for M. tuberculosis, so for testing impact of efflux on antimicrobial activity, half or a quarter of this could be used without affecting growth. In fact, verapamil has been used at 88µM in two publications from Parish group (PMID: 22123255, 18489786) and in one from Nuermberger lab (PMID: 2718580), at 110µM in papers from Bishai (PMID: 24126586) and Sterling (PMID: 27261264) labs, at 165 µM in a publication from Ainsa lab (PMID: 29987141), and up to 170 µM in papers from Ramakrishnan lab (for example, PMID: 24532601).

Anyway, the current manuscript should address such discrepancy satisfactorily, either by removing all data related with verapamil (hence testing efflux only by using other inhibitors such as reserpine and CCCP), or by introducing a paragraph commenting at least all data shown above.

7. PLOS authors have the option to publish the peer review history of their article (what does this mean?). If published, this will include your full peer review and any attached files.

Reviewer #1: No

---

## [Author Response · Author response to Decision Letter 1]

25 Nov 2020

Reviewer #1: 

As noted by the reviewer, the concentration of verapamil we used was lower than in other studies. We were cautious in making sure there was no inhibition of growth at all (%inhibition ,20) and we had seen variability between batches from suppliers. We have removed the conclusion from the abstract and added a caveat regarding this low concentration and that it may not be sufficient to inhibit efflux.

---

## [Decision Letter · Decision Letter 2]

8 Dec 2020

Phenoxyalkylimidazoles with an oxadiazole moiety are subject to efflux in Mycobacterium tuberculosis

PONE-D-20-28352R2

Dear Dr. Parish,

We’re pleased to inform you that your manuscript has been judged scientifically suitable for publication and will be formally accepted for publication once it meets all outstanding technical requirements.

Kind regards,

Olivier Neyrolles

Section Editor

PLOS ONE

Reviewers' comments:

Reviewer's Responses to Questions

**Comments to the Author**

1. If the authors have adequately addressed your comments raised in a previous round of review and you feel that this manuscript is now acceptable for publication, you may indicate that here to bypass the “Comments to the Author” section, enter your conflict of interest statement in the “Confidential to Editor” section, and submit your "Accept" recommendation.

Reviewer #1: All comments have been addressed

2. Is the manuscript technically sound, and do the data support the conclusions?

Reviewer #1: Yes

3. Has the statistical analysis been performed appropriately and rigorously? 

Reviewer #1: N/A

4. Have the authors made all data underlying the findings in their manuscript fully available?

Reviewer #1: Yes

5. Is the manuscript presented in an intelligible fashion and written in standard English?

Reviewer #1: Yes

6. Review Comments to the Author

Reviewer #1: The authors have now addressed satisfactorily all points raised in previous versions of the manuscript.

7. PLOS authors have the option to publish the peer review history of their article (what does this mean?). If published, this will include your full peer review and any attached files.

Reviewer #1: No

---

## [Editor Report · Acceptance letter]

21 Dec 2020

PONE-D-20-28352R2 

Phenoxyalkylimidazoles with an oxadiazole moiety are subject to efflux in *Mycobacterium tuberculosis*

Dear Dr. Parish:

I'm pleased to inform you that your manuscript has been deemed suitable for publication in PLOS ONE. Congratulations! Your manuscript is now with our production department. 

Kind regards, 

on behalf of

Dr. Olivier Neyrolles 

Section Editor

PLOS ONE